# Unmet Needs in the Assessment of Right Ventricular Function for Severe Tricuspid Regurgitation

**DOI:** 10.3390/diagnostics13182885

**Published:** 2023-09-08

**Authors:** Vasileios Anastasiou, Maria-Anna Bazmpani, Stylianos Daios, Dimitrios V. Moysidis, Thomas Zegkos, Matthaios Didagelos, Theodoros Karamitsos, Konstantinos Toutouzas, Antonios Ziakas, Vasileios Kamperidis

**Affiliations:** 1First Department of Cardiology, AHEPA Hospital, School of Medicine, Aristotle University of Thessaloniki, 541 24 Thessaloniki, Greece; vasianas44@gmail.com (V.A.); mariannabaz@hotmail.gr (M.-A.B.); stylianoschrys.daios@gmail.com (S.D.); dimoysidis@gmail.com (D.V.M.); zegkosth@gmail.com (T.Z.); manthosdid@yahoo.gr (M.D.); karamits@gmail.com (T.K.); tonyziakas@hotmail.com (A.Z.); 21st Department of Cardiology, Hippokration Hospital, School of Medicine, National and Kapodistrian University of Athens, 157 72 Athens, Greece; ktoutouz@gmail.com

**Keywords:** tricuspid regurgitation, right ventricular function, multimodality imaging, echocardiography, cardiac computed tomography, cardiac magnetic resonance

## Abstract

Tricuspid regurgitation (TR) is a highly prevalent valvular heart disease that has been long overlooked, but lately its independent association with adverse cardiovascular outcomes was recognized. The time point to intervene and repair the tricuspid valve is defined by the right ventricular (RV) dilation and dysfunction that comes up at a later stage. While guidelines favor tricuspid valve repair before severe RV dysfunction ensues, the definition of RV dysfunction in a universal manner remains vague. As a result, the candidates for transcatheter or surgical TR procedures are often referred late, when advanced RV dysfunction is established, and any derived procedural survival benefit is attenuated. Thus, it is of paramount importance to establish a universal means of RV function assessment in patients with TR. Conventional echocardiographic indices of RV function routinely applied have fundamental flaws that limit the precise characterization of RV performance. More recently, novel echocardiographic indices such as strain via speckle-tracking have emerged, demonstrating promising results in the identification of early RV damage. Additionally, evidence of the role of alternative imaging modalities such as cardiac computed tomography and cardiac magnetic resonance, for RV functional assessment in TR, has recently arisen. This review provides a systematic appraisal of traditional and novel multimodality indices of RV function in severe TR and aims to refine RV function assessment, designate future directions, and ultimately, to improve the outcome of patients suffering from severe TR.

## 1. Introduction

Clinically significant tricuspid regurgitation (TR) is a common valvular heart disease affecting 0.55% of the general population, with functional TR accounting for the most severe TR cases [1,2]. Despite its rising prevalence, TR has been long overlooked, being largely considered the sequela of advanced left sided heart disease, rather than a valvular entity of independent prognostic relevance. Contemporary data have shed light to its concealed, independent association with excess mortality, outlining the need for drastic therapies [3,4,5]. Although surgical intervention was until recently considered the only interventional approach, transcatheter tricuspid valve repair has emerged as an alternative treatment strategy for inoperable symptomatic patients [6].

Significant TR has a deleterious impact on right ventricular (RV) size and function; TR will progressively induce tricuspid annular dilation and tricuspid leaflet tethering due to RV remodeling, while retaining normal RV function at the compensating phase. When the point of RV adaptation to volume overload is surpassed, patients suffer RV functional compromise and enter a vicious cycle of recurrent heart failure admissions, further aggravating their prognosis (Figure 1) [7]. In this regard, a well-timed selection of candidates for interventional TR treatments in advance of the RV failure is of paramount importance, while at the same time remaining a difficult task. The ESC and AHA/ACC guidelines recommend that the intervention for severe TR treatment takes place before RV dysfunction is established; however, they remain vague in terms of practically defining RV dysfunction [8,9].

The current review aims to enlighten the assessment of RV dysfunction in the presence of severe TR by addressing the merits of novel imaging modalities in echocardiography, cardiac computed tomography (CCT), and cardiac magnetic resonance (CMR). It ultimately endeavors to provide a guide for multiparametric assessment of RV function that will enable well-timed therapeutic decision making in TR.

## 2. Echocardiographic Assessment of Right Ventricular Function

### 2.1. Conventional RV Function Indices

Challenges in the evaluation of RV performance stem mostly from its complex anatomy. The RV is a crescent-shaped structure including an outlet, apex, and inlet portion. Hence, incorporating the contractile force of all three components with the measurement of one single index is practically impossible [10]. Conventionally, echocardiography is the mainstay imaging modality for evaluating RV function in the presence of significant TR, owing to its ease of use, safety, availability, and lower costs. Numerous traditional indices of RV systolic function in severe TR have been embraced in clinical practice due to their practicality, despite having fundamental flaws [11].

Tricuspid annular plane systolic excursion (TAPSE) evaluates the distance of systolic excursion of the RV annular segment along its longitudinal plane, while S’ measures the peak systolic annular velocity via Doppler tissue imaging at the tricuspid annulus level. Both are widely established indices, recommended for RV function assessment with a suggested cutoff value of 17 mm and 9.5 cm/s, respectively [11]. In contrast to the aforementioned indices which describe solely the longitudinal contractile force at the tricuspid annular level, fractional area change (FAC) additionally accounts for the circumferential RV contraction. FAC is defined as (end-diastolic area–end-systolic area)/end-diastolic area ×100 and has shown a fair correlation with RV ejection fraction (RVEF) via CMR [12]. Its abnormality cutoff is set at 35% for the general population as dictated by the respective guidelines [11].

Despite their widespread use, evidence of the prognostic value of those indices is contradictory in severe TR. In the study by Karam et al., which included 249 high-risk patients with severe TR undergoing transcatheter repair, neither TAPSE nor FAC assessed pre-intervention could elicit long-term prognostic information [13]. Conversely, for conservatively managed patients, conventional RV function indices offered prognostic information. In a large registry of 1298 patients with significant secondary TR who were medically managed, subjects with TAPSE < 17 mm suffered the worst survival regardless of their RV dimensions [14]. The discrepancy of findings between the two studies might be attributable to the large sample size of the latter, where even the simplified TAPSE might still be indicative of the natural history of RV failure. 

Kwon et al. associated reduced preoperative S’ with unfavorable outcomes in patients with severe functional TR, with previous mitral valve surgery having isolated tricuspid valve surgery, but their study was small and underpowered [15]. For surgically managed patients, more contemporary data were obtained by Dreyfus et al. who examined the post-operative incidence of in-hospital mortality in patients with primary or secondary symptomatic severe TR undergoing isolated tricuspid valve surgery. This study demonstrated that moderate/severe RV dysfunction, as assessed via TAPSE, S’, or FAC was independently associated with in-hospital mortality [16]. Similar results were delivered by Subbotina et al., who identified reduced TAPSE as a risk factor for early post-operative mortality in a TR cohort undergoing isolated or combined tricuspid valve surgery [17]. In keeping with those findings, Algarni et al. examined 548 patients with secondary TR undergoing tricuspid valve repair concomitant with left-side valve surgery and identified a pre-operative TAPSE < 15 mm as an independent predictor of long-term mortality [18]. In terms of post-operative RV function assessment, an RV FAC value of ≥31% has been shown to predict event-free survival with a sensitivity of 90% and a specificity of 83% after isolated tricuspid valve surgery [19].

TAPSE and S’ are simplified estimates of RV function and subjects to significant limitations in the setting of severe TR. Both indices are angle and load dependent and they assume that the displacement of a single point of the tricuspid annulus represents the overall function of the complex, three-dimensional structure of the RV [11]. Therefore, they exhibit only modest a correlation with the volumetric gold standard of RV EF via CMR [20]. Moreover, the movement of tricuspid annulus is usually accentuated in severe TR, resulting in overestimation of RV performance [11]. Even after tricuspid valve surgery TAPSE can be inaccurate and lead to underestimation of RV systolic function [21]. With respect to FAC, it is measured from a single acquisition of the RV from an apical four-chamber view; hence, it is limited by appreciable geometrical assumptions and demonstrates only fair inter-observer variability [22]. Main studies addressing the prognostic role of those three indices in the setting of severe TR are summarized in Table 1.

### 2.2. Three-Dimensional Echocardiography

Left ventricular EF by two-dimensional Simpson’s biplane method is the main echocardiographic parameter implemented to evaluate ventricular function for left-sided valvular heart disease in clinical practice. Simpson’s biplane method cannot be applied for the RV, which can only be assessed from a single plane in an apical RV-focused view via two-dimensional echocardiography. However, echocardiography allows the estimation of RV EF with the use of two-dimensional imaging that allows real RV full-volume capture and it is not dependent on the image plane orientation of the two-dimensional imaging.

Three-dimensional echocardiography has shown less volume underestimation than two-dimensional echocardiography and acceptable performance for volume quantification, comparable to CMR [25]. A head-to-head comparison between conventional RV function indices and three-dimensional RV EF demonstrated superior association of the latter in outcomes [26]. Among all other measurements of RV function, three-dimensional RVEF is the only index that allows assessment of all regions of the RV as well as the tricuspid valve leaflets [27,28]. In a cohort of 75 patients with severe TR undergoing transcatheter repair, preoperative three-dimensional RV EF predicted 1-year mortality, whereas TAPSE and FAC did not [29]. Moreover, higher pre-operative three-dimensional RV EF was associated with more pronounced improvement of functional capacity after the procedure [29].

Despite the benefits of three-dimensional echocardiography, the use of RV EF can markedly overestimate RV function for patients with severe TR, as it does not account for the relationship between contractility and afterload, but only describes the percentage of volume change per cardiac cycle [30]. Three-dimensional RV EF is an accurate expression of volumetric RV changes but fails to account for the direction of blood flow; forward to the pulmonary artery or backwards to the right atrium. In the case of severe TR, the RV largely empties in the low-pressure right atrium through the regurgitant jet, and this may mask a reduced RV inotropic state. Thus, the RV EF may not demonstrate an impaired RV myocardial performance since the volumetric change reflects both the forward flow and the amount of regurgitant volume and not active myocardial shortening per se. These limitations have stimulated the development and application of novel indices such as strain imaging for the RV.

### 2.3. Longitudinal Strain

In patients with severe TR presenting with RV volume overload, the contraction of subendocardial longitudinal fibers might be the first to be insulted indicating an early phase of ventricular damage, while contractility of the mid-circumferential RV layers can still be preserved or even augmented to compensate for the loss of longitudinal force (Figure 1) [24]. When circumferential contraction is lost, this represents an advanced stage of RV failure, and any attempt to intervene and change the natural history of TR might come with little clinical benefit [24]. Hence, appropriate assessment of RV longitudinal contraction is of paramount importance and might be the most pertinent to set optimal cutoff values to indicate the best time for intervention.

For over two decades, two-dimensional speckle-tracking echocardiography has emerged as a novel imaging technique assessing intrinsic myocardial function. Despite being initially utilized for the left ventricle, its applicability has recently been extended to the RV. RV global longitudinal strain (RVGLS) and RV free-wall longitudinal strain (RVFWLS) are the two main indices of speckle-tracking echocardiography for the assessment of RV performance, with a strong correlation to CMR-derived RV EF [31]. These indices offer substantial benefits over traditional indices of RV longitudinal function such as TAPSE or S’, as they are less dependent on geometry, cardiac translational motion, and loading conditions [22]. Since they follow the movement of myocardial speckles across the whole range of myocardial wall, they more accurately reflect more the overall RV performance [22]. However, a considerable limitation is the absence of unified, robust normality threshold both for RV GLS and RV FWLS [32].

RV longitudinal strain has provided evidence of refined risk stratification for TR patients. Prihadi et al. studied a large cohort of 896 patients with moderate or severe functional TR and disclosed that RV FWLS identified a higher number of patients with abnormal RV function compared to the traditional TAPSE and FAC [33]. This observation denotes the superior sensitivity of this index to capture subtle myocardial damage compared to the conventional echocardiography parameters of RV function [33]. In line with those findings, in a group of severe TR patients mainly functional in etiology, Ancona et al. showed that RV FWLS reclassified almost 50% of patients with apparently normal RV function, via TAPSE, FAC, or S’, to impaired RV function [34].

Apart from the ability to detect RV dysfunction in advance of the conventional parameters, RV longitudinal strain provides prognostic information for patients with severe TR treated conservatively. Prihadi et al. demonstrated that RV FWLS is associated, over and above TAPSE and FAC, with long-term outcomes [33]. In the study by Ancona et al., an optimal cutoff value of RV FWLS of −14% was identified to predict all-cause mortality [34]. Similar findings were published by Hinojan et al., who demonstrated superiority of RV GLS and RV FWLS to predict outcomes compared to conventional RV parameters in a cohort of secondary, at least severe TR patients without indication for intervention, implying that speckle-tracking echocardiography might be best suited to inform regarding intervention [35]. Downstream RV impairment will affect right atrial function, and subjects with a phenotype of impaired RV and right atrial strain are at higher risk for events [36].

RV strain could serve as a tool to select appropriate surgical candidates for TR repair at an early stage of the disease, before irreversible RV damage ensues, to optimize outcomes. Kim et al. indicated in a cohort of 115 patients with severe functional TR undergoing isolated tricuspid valve surgery that a pre-operative RV FWLS > −24% could independently predict dismal post-operative outcomes [37]. Furthermore RV strain provided incremental information on top of baseline clinical characteristics, whereas RV end-systolic area and RV FAC did not [37]. Similarly, post-operatively impaired RV GLS was independently associated with poor outcomes, highlighting the prognostic role of post-operative RV failure [38].

RV longitudinal strain provides information on the myocardial contraction beyond a mere description of volumetric RV changes. As such, it appears to be superior to other RV indices and could potentially refine risk stratification and optimal thresholds for intervention in TR. Figure 2 illustrates the progression from conventional RV indices to contemporary RV strain imaging via echocardiography in three diverse phenotypes of TR. A summary of studies addressing the prognostic value of RV strain in TR is shown in Table 2.

### 2.4. Right Ventricular-Pulmonary Arterial Coupling

Although RV function assessment is paramount in severe TR, it fails to account for the subtending RV afterload, which may vary substantially between patients. Besides RV dysfunction, the prognostic role of pulmonary hypertension, which is a common hemodynamic complication of long-standing heart failure and TR, has been recognized [42]. RV-pulmonary arterial (PA) coupling has emerged as a novel, comprehensive index that allows the evaluation of RV function in relation to the underlying RV afterload [43]. This index can be readily assessed non-invasively through the ratio of two standard echocardiographic measurements: TAPSE over pulmonary artery systolic pressure (PASP). RV FWLS has also been used instead of TAPSE, but currently, there is less evidence available [44].

TAPSE/PASP ratio has demonstrated good correlation with the invasive pressure-volume loop-derived end-systolic/arterial elastance, which is the gold-standard measure of RV-PA coupling [45]. This index was initially applied as an outcome predictor for heart failure patients [43,46] but has recently been expanded to valvular heart disease [44,47,48], including patients with severe TR [49]. Fortuni et al. disclosed in a cohort of 1149 patients with at least moderate functional TR treated mostly conservatively, that RV-PA uncoupling, as defined via TAPSE/PASP < 0.31 mm/mmHg, was associated with significantly reduced survival [49]. Brener et al. expanded on those findings by demonstrating that impaired TAPSE/PASP was an independent predictor of poor post-operative outcomes for patients with severe TR undergoing transcatheter tricuspid valve repair [44]. More recently, a single-center study by Ancona et al. indicated that RV FWLS/PASP might be a superior prognosticator than TAPSE/PASP in severe TR, with a threshold of 0.26%/mmHg predicting outcomes [50].

While TAPSE/PASP could play a role to select appropriate candidates for percunatenous tricuspid valve interventions with severe TR, its utility is questionable for cases with greater than severe (massive or torrential) TR. In such cases, TAPSE/PASP may overestimate the true coupling of the RV-PA circuit, as echocardiographic PASP is underestimated and correlates poorly with the respective invasive PASPs. In a cohort of 126 patients with greater than severe TR undergoing transcatheter repair, only the TAPSE/PASP using the invasive PASP could provide prognostic information, whereas the echocardiographic TAPSE/PASP could not [51].

## 3. Assessment of Right Ventricular Function via Cardiac CT

### Right Ventricular Ejection Fraction

CCT use in patients with significant TR is advocated for use in pre-procedural planning of the transcatheter repair. CCT has a high spatial resolution and provides three-dimensional cardiac data. This enables the precise assessment of the tricuspid annular morphology and the sub valvular anatomy, the characterization of surrounding structures, the determination of pacing lead location, the definition of optimal fluoroscopic angulations, and vascular access root planning [52].

Beyond the value of CCT for anatomical assessment, its role in RV function evaluation has recently been explored. Due to its optimal spatial resolution, CCT enables excellent endocardial–blood pool interface definition [53] and concomitantly enables the acquisition of high-resolution, three-dimensional images throughout the whole cardiac cycle [54]. Hence, RV EF can be easily estimated from the three-dimensional end-diastolic and end-systolic volume. RV volumes and RV EF by CCT have demonstrated excellent correlation with the respective CMR-derived measures [55]. In a cohort of symptomatic patients with severe TR undergoing transcatheter tricuspid valve repair, Tanaka et al. utilized CCT data to derive RV EF readings from short-axis tracings and reported an excellent feasibility of 94% [56]. CCT-derived RV EF has emerged as a potent predictor of 1-year composite outcomes, and it demonstrated incremental prognostic information over standard echocardiographic measurements of RV function [56]. Interestingly, intermodal agreement between CCT-derived RV EF and echocardiographic RV dysfunction was poor [56].

Currently, CCT is not implemented in everyday clinical practice for the RV EF evaluation, due to a lack of availability, radiation exposure, and the need for an iodinated contrast, making the available data scarce. However, RV EF assessment should be encouraged in all the patients with TR undergoing a CCT for any reason, such as coronary artery disease evaluation or pre-procedural planning of valvular repair or replacement.

## 4. Assessment of Right Ventricular Function by CMR

### 4.1. Right Ventricular Ejection Fraction

CMR is considered the gold-standard modality for volumetric quantification of the RV. CMR derived RV EF bypasses any potential geometrical assumptions of the RV shape and is based on the real RV volumes in end-diastole and end-systole [57]. This index is considered the gold standard of RV function, as it incorporates information both on the longitudinal and circumferential contraction of the RV, allows for optimal endocardial border discrimination, and is very reproducible [58]. To obtain RV volumes for RV EF, a short-axis stack of images is utilized from the left ventricular scout to ensure that all RV portions from the base to the apex are covered, with a slice thickness of 6–8 mm and a 2–4 mm interslice gap (Figure 3) [59].

Park et al. identified RV EF as an independent predictor of cardiac death and major postoperative cardiac events in 75 consecutive patients undergoing corrective surgery for severe functional TR [60]. In a cohort of 79 patients scheduled for transcatheter tricuspid valve repair for severe TR, CMR-derived RV EF < 45% emerged as an independent predictor of outcome for the composite endpoint of heart failure hospitalization and all-cause mortality [24]. Notably, patients with reduced CMR RV EF and echocardiographic TAPSE < 17 mm had reduced longitudinal, radial, and circumferential strain as determined with feature-tracking CMR and exhibited worse survival compared to patients with reduced TAPSE but preserved CMR-RV EF [24]. This finding indicates that RV EF impairment denotes an advanced stage of RV dysfunction in the natural history of TR, and intervention should potentially be pursued at an earlier stage. More contemporary CMR data of patients with severe TR have indicated that effective RV EF, which corrects for the underlying regurgitant volume might be a stronger outcome predictor compared to RV EF [61]. This index is based on the forward stroke volume of the RV and is derived from the equation (net pulmonary flow)/(RV-EDV) [61].

Despite its excellent accuracy in volume quantification, CMR-derived RV EF still relies on a mere description of RV volumetric changes in a cardiac cycle, potentially overestimating RV function for patients with a severe TR background. Even the effective RV EF that accounts for the regurgitant volume does not providing any information about the myocardial RV function per se, and is considered to be impaired in an advanced stage of RV dysfunction. However, earlier signs of RV decompensation might be of stronger clinical relevance to drive the interventional decisions on TR.

### 4.2. Longitudinal Strain 

Most available evidence of longitudinal strain in TR stems from speckle-tracking echocardiography which, however, is limited by the two-dimensional nature of this modality for a complex three-dimensional structure. Recently, CMR has allowed the estimation of strain using standard cine images, without the need for separate pulse sequences [62]. A multicenter study comprising 544 patients with severe, functional TR addressed the role of RV FWLS via feature tracking CMR [41]. Not only was RV FWLS an independent predictor of death, but it additionally demonstrated incremental prognostic value when added on top of clinical and imaging variables including RV EF [41]. Nonetheless, further evidence is warranted on the role and optimal cutoff values of strain imaging via CMR before routine clinical application is feasible. A clinical case of feature tracking longitudinal strain for RV assessment via CMR is depicted in Figure 3. A summary of studies addressing the prognostic value of RV strain in TR is shown in Table 2.

### 4.3. Tissue Characterization

A unique advantage of CMR is the myocardial tissue characterization which has been studied for the left ventricle in cases of left-sided valve disease [63,64], but remains unexplored for the RV in the case of TR. Challenges in applying tissue characterization sequences for the RV are related to its thin wall. Normally, the RV free wall does not exceed 3–5 mm, which limits spatial resolution [65]. However, RV myocardial tissue in significant TR is subject to increased pressure and volume load that could lead to increased wall thickness and detectable fibrotic changes similar to the left ventricle [66]. In this regard, applying gadolinium enhancement and extracellular volume mapping techniques for the RV could provide more information on RV dysfunction and help implement early TR intervention.

## 5. The Role of Biomarkers

In an attempt to monitor the trajectory of severe TR and apply timely intervention, the approach to RV function assessment should be holistic. The assessment of imaging indices should be complemented by biochemical biomarkers in clinical practice. Elevated B-type natriuretic peptide levels before intervention may connote advanced disease and have been associated with poor post-operative course in severe TR [67]. Overall, increased natriuretic peptides predict poor outcome in primary and secondary TR and right heart failure [68].

In patients with severe TR and right heart failure, physical signs of overt tissue oedema resulting from long-standing intravascular and interstitial fluid volume expansion is the prevailing symptom. Carbohydrate antigen 125 is a reliable surrogate of congestion in heart failure and is independent of renal function, age, and weight. Particularly in the setting of severe TR, carbohydrate antigen 125 has demonstrated stronger association with mortality compared to natriuretic peptides, which may not reflect the contribution of functional TR towards the disease pathophysiology [69]. Moreover, carbohydrate antigen 125 has shown independent association with TR severity in a large cohort with acute heart failure [70]. Therefore, its routine evaluation might be better suited in severe TR to provide further insight of the disease severity.

## 6. Author’s Perspectives

Early intervention for left-sided valve diseases to prevent irreversible left ventricular damage is an established practice applied in everyday clinical practice. However, a long way has to be made to achieve an early intervention on the tricuspid valve in case of TR to avoid RV dysfunction. The high operative morbidity and mortality documented for the surgical treatment of TR is largely attributable to the lack of universally established indices of RV dysfunction and accepted cutoff values, leading to inconsistent practices regarding the right time for intervention [71]. This miscommunication drives late and futile tricuspid interventions in the context of advanced, irreversible RV damage and, hence, poor outcomes. 

According to the ESC and ACC/AHA guidelines, severe TR should be treated interventionally/surgically when RV failure symptoms arise, and for the asymptomatic patients intervention is recommended in the presence of progressive RV dilatation or systolic dysfunction. No recommendations are available for patients with severe asymptomatic functional TR [8,9]. However, the definition of RV dysfunction can be multifaceted and the best parameter to establish RV dysfunction is not defined, while a strategy of waiting to manifest severe RV dysfunction or RV failure symptoms to decide on intervention seems suboptimal. 

Promising evidence has emerged on the role of strain imaging via speckle-tracking echocardiography, which is an index of early RV damage associated with outcome in asymptomatic patients with severe TR [33,35,37,38]. Strain imaging can detect subtle intrinsic myocardial dysfunction in advance of the RV EF deterioration, and its impairment is considered as an early indirect expression of myocardial fibrosis and myocardial damage. Thus, patients with impaired longitudinal RV stain who still retain normal indices of circumferential contractility such as RV EF can be those who benefit the most of a tricuspid valve intervention. It should be highlighted that both the echocardiographic three-dimensional RV EF and the CMR RV EF, which is considered the gold standard of RV function, are impaired at an advanced stage of the severe TR disease.

In the current era of the new TR classification and the accompanied phenotypes of RV remodeling, the implementation of different cutoff values for RV dysfunction might be necessary [72]. For instance, patients with atrial functional TR predominantly exhibit right atrial and tricuspid annular dilatation and less RV remodeling, and less pronounced tenting forces and more preserved indices of RV function compared to ventricular functional TR [73]. Therefore, a tailored approach to define disease severity and RV dysfunction is necessitated based on the type of TR (Figure 2). Further research should be performed to delineate how RV dysfunction should be defined in the setting of diverse TR types (organic, functional ventricular, functional atrial, and implantable-electronic-device-related) by applying all the available modalities and methods, with a special focus on strain imaging, which appears to be the leading expression of prompt RV dysfunction.

## 7. Conclusions

Since RV function should dictate the timing of tricuspid valve repair in severe TR, it has to be assessed in a universal, easily accessible, and widely available manner that can detect even subtle RV dysfunction. RV strain imaging seems to be the key in revealing in advance the anticipated RV decompensation. RV longitudinal strain via speckle-tracking echocardiography has considerable strengths compared to traditional simplified parameters of RV function in describing early RV damage. Moreover, CMR-derived feature-tracking strain has arisen, showing additive value over the gold-standard measure of RV EF. Although evidence is promising, further prospective research is crucial before strain-driven patient management in TR can be advocated.

## Figures and Tables

**Figure 1 diagnostics-13-02885-f001:**
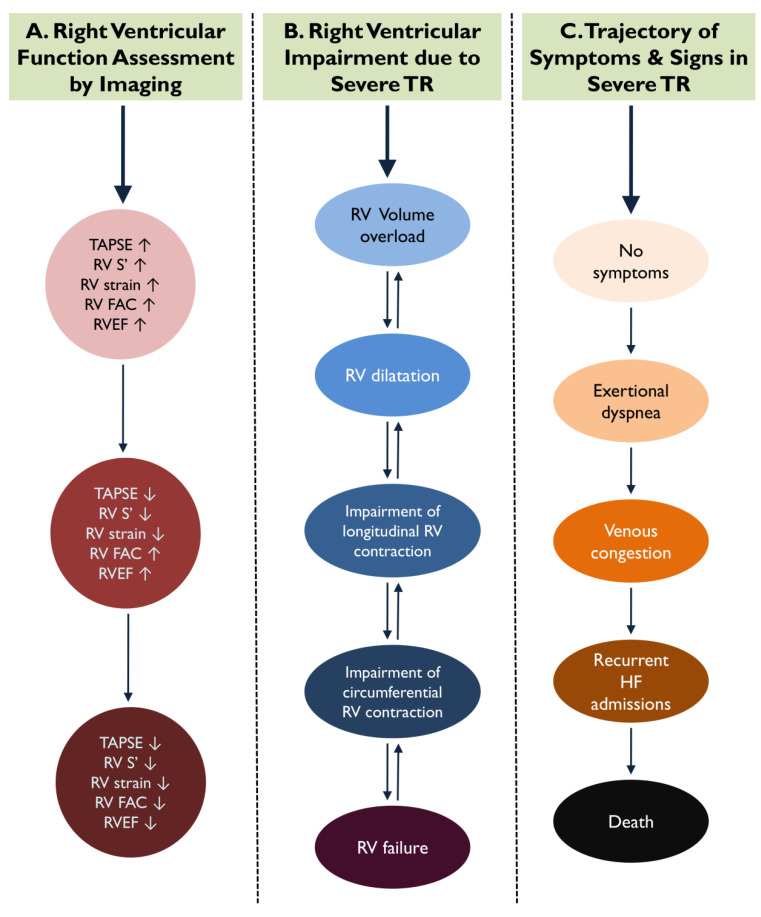
During the course of significant tricuspid regurgitation (TR), indices of longitudinal right ventricular (RV) function are impaired earlier compared to parameters of circumferential contraction. This figure demonstrates the parallel trajectories of RV dysfunction via imaging parameters (**A**), RV adverse remodeling (**B**), and the onset of RV failure symptoms and signs (**C**) in severe TR.

**Figure 2 diagnostics-13-02885-f002:**
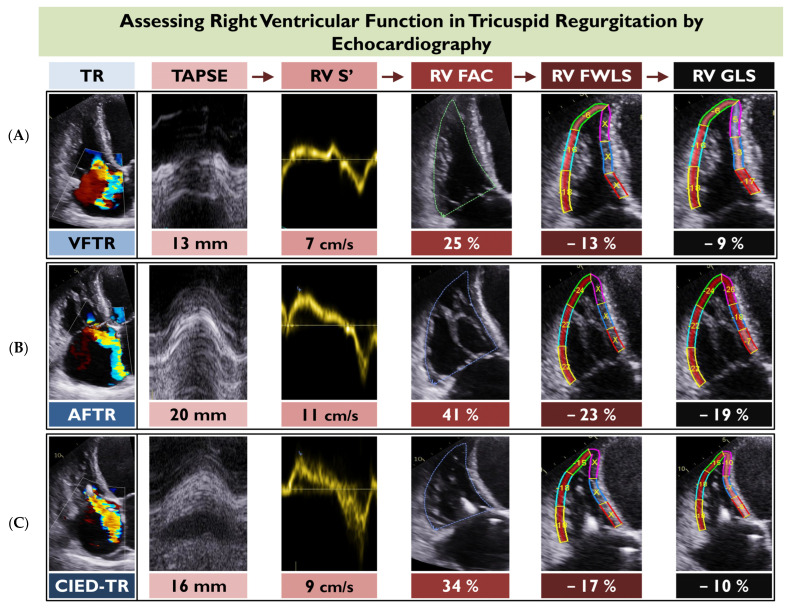
This figure depicts the application of the main two-dimensional echocardiographic indices of right ventricular (RV) function for three different case examples of tricuspid regurgitation, including ventricular functional TR (VFTR) (**A**), atrial functional TR (AFTR) (**B**), and cardiovascular implantable electronic device-related TR (CIED-TR) (**C**). Of note, all indices of RV function appear more preserved for AFTR compared to the other two TR types. FAC, fractional area change; RV FWLS, right ventricular free wall longitudinal strain; RV GLS; right ventricular global longitudinal strain; TAPSE; tricuspid annular systolic plane excursion.

**Figure 3 diagnostics-13-02885-f003:**
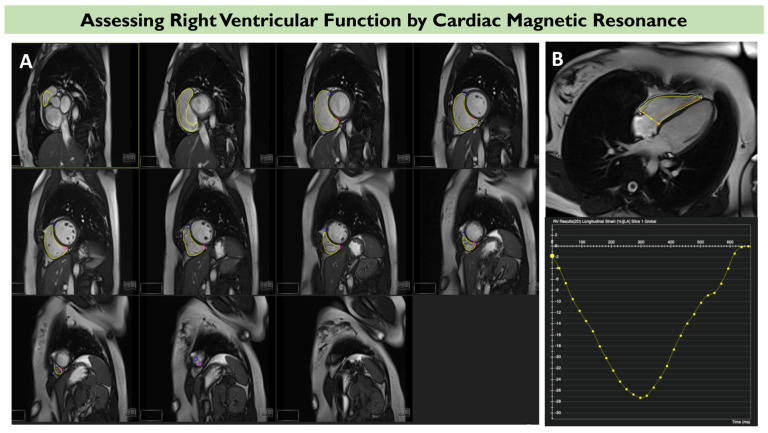
An example right ventricular (RV) function assessment via cardiac magnetic resonance (CMR). Volumetric assessment of RV function is illustrated in panel (**A**), with contouring of the RV from base to apex at end diastole (yellow traces). Panel (**B**) shows an example of CMR feature tracking of the right ventricle, where endocardial and epicardial contours are manually traced in the four-chamber view (upper panel). The corresponding global longitudinal strain curves are automatically constructed using software CVI42 (lower panel).

**Table 1 diagnostics-13-02885-t001:** Summary of important studies addressing the prognostic role of conventional echocardiographic right ventricular function indices in tricuspid regurgitation.

Author	Year	Number of Patients	Patient Population	Method of RV Function Assessment	Primary Outcome	Key Findings
Kwon et al. [15]	2006	18	Severe functional TR after mitral surgery undergoing isolated TV surgery	S’	Unfavorable postoperative clinical outcomes	Patients with S′ < 9.5 cm/s had unfavorable postoperative clinical outcomes, while patients with S’ > 13 cm/s had favorable outcomes
Park et al. [19]	2011	69	Severe TR undergoing isolated TV surgery	FAC	Operative mortality, cardiovascular death, repeated surgery and readmission	Early postoperative RV-FAC predicted long-term event-free survival RV FAC ≥ 31% predicted event-free survival with a sensitivity of 90% and a specificity of 83%
Subbotina et al. [17]	2017	191	Severe TR undergoing isolated or combined TV surgery	TAPSE	30-day survival after surgery	Preoperative RV dysfunction was identified as risk factor for early mortality
Dietz et al. [14]	2019	1292	Moderate and severe secondary TR	TAPSE	All-cause mortality	Patients with TAPSE < 17 mm had significantly worse survival than patients with normal TAPSE > 17 mm RV systolic dysfunction showed the lowest survival regardless of the RV dimensions
Suh et al. [23]	2019	100	Severe TR undergoing TV surgery	TAPSE, S’, FAC	Postoperative RV dysfunction	Impairment of at least two RV parameters on preoperative echo was associated with postoperative RV dysfunction
Karam et al. [13]	2020	249	Severe symptomatic TR undergoing TTVR	TAPSE, FAC	Hospitalization for heart failure	No difference in outcomes according to baseline TAPSE No difference in outcomes according to baseline FAC
Kresoja et al. [24]	2021	79	Symptomatic, significant TR undergoing TTVR	TAPSE	All-cause mortality or first heart failure hospitalization	TAPSE was not able to predict the composite outcome
Algarni et al. [18]	2021	548	Severe secondary TR undergoing TV repair with left-side valve surgery	TAPSE	Long-term mortality	TAPSE < 15 mm was independently associated with poor long-term survival
Dreyfus et al. [16]	2022	466	Severe symptomatic TR undergoing isolated TV surgery	TAPSE, S’, FAC	In-hospital mortality	Moderate/severe RV dysfunction was independently associated with in-hospital mortality

FAC, fractional area change; RV, right ventricle; TAPSE, tricuspid annular plane systolic excursion; TR, tricuspid regurgitation; TTVR; transcatheter tricuspid valve repair; TV, tricuspid valve.

**Table 2 diagnostics-13-02885-t002:** Summary of studies addressing the prognostic role of right ventricular strain imaging in tricuspid regurgitation.

Author	Year of Publication	Number of Patients	TR Population	RV Strain Cut-Off	Primary Outcome	Key Findings
Echocardiography						
Prihadi et al. [33]	2019	896	Moderate and severe functional TR	−23%	All-cause mortality	RV FWLS identifies higher rates of RV dysfunction compared to TAPSE and FAC In significant FTR, RV FWGLS was an independent predictor of mortality
Ancona et al. [34]	2021	250	Severe TR	−14%	All-cause mortality	RV FWLS < 14% is an independent predictor of mortality RV FWLS < 17% is independently associated with clinical RVHF
Bannehr et al. [39]	2021	1089	Mild, moderate, and severe TR	−18%	All-cause mortality	RV strain < 18% was an independent predictor of 2-year all-cause mortality
Wang et al. [40]	2021	262	Moderate-severe isolated TR	−11%	All-cause mortality	Reduced RV FWLS is associated with higher risk for all cause death
Kim et al. [37]	2021	115	Severe functional TR undergoing isolated TV surgery	−24%	Composite of cardiac death and unplanned readmission	RV FWLS < 24% was an independent predictor of outcome in patients undergoing surgery for severe FTR.
Kim et al. [38]	2022	111	Severe TR undergoing isolated TV surgery	−17.2%	Cardiac death, HFH, redo TV surgery, heart transplantation	Preoperative RV GLS < 17.2% was a predictor of poor prognosis after isolated TV surgery
Vely et al. [36]	2022	241	At least severe TR	-	All-cause mortality or HFH	Patients with severely dilated heart chambers and impaired RV and right atrial strain had the worst outcome
Hinojar et al. [35]	2023	151	At least severe secondary TR without indication for intervention	-	All-cause mortality or HFH	RV FWLS was independently associated with mortality and heart failure, whereas conventional indices were not
Cardiac Magnetic Resonance						
Romano et al. [41]	2021	544	Clinically reported severe FTR	−16%	All-cause mortality	RV FWLS < 16% is an independent predictor of mortality

HFH, heart failure hospitalization; FAC, fractional area change; FTR, functional tricuspid regurgitation; RVFWLS, right ventricular free wall global longitudinal strain; RV GLS, right ventricular global longitudinal strain; RVHF, right ventricular heart failure; TAPSE, tricuspid annular plane systolic excursion; TR, tricuspid regurgitation; TV, tricuspid valve.

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
