# Peer review of "Unmet Needs in the Assessment of Right Ventricular Function for Severe Tricuspid Regurgitation"

_diagnostics, 2023, doi:10.3390/diagnostics13182885_

Round 1

Reviewer 1 Report

·      The paper does not explain clearly its advantages with respect to the literature: it is not clear what is the novelty and contributions of the proposed work: does it propose a new method? Or does the novelty only consists in the application? Related works should be discussed: https://doi.org/10.1016/j.inffus.2022.11.004; https://doi.org/10.1152/ajpheart.00416.2020

·      The advantage of the proposed method with respect to other methods in the literature should be clarified.

·      The paper does not provide significant experimental details needed to correctly assess its contribution: What is the validation procedure used?

·      Quality of figures is so important too. Please provide some high-resolution figures. Some figures have a poor resolution

·      Conclusion should state scope for future work.

·      Results need more explanations. Additional analysis is required at each experiment to show the its main purpose.

·      Need detailed explanation of the preprocessing steps.

none

Author Response

Manuscript ID: diagnostics-2514341

Type of manuscript: Review

Title: Unmet Needs in the Assessment of Right Ventricular Function for
Severe Tricuspid Regurgitation

Response to Reviewer 1 Comments

Point 1: The paper does not explain clearly its advantages with respect to the literature: it is not clear what is the novelty and contributions of the proposed work: does it propose a new method? Or does the novelty only consists in the application? Related works should be discussed: https://doi.org/10.1016/j.inffus.2022.11.004; https://doi.org/10.1152/ajpheart.00416.2020

Response 1: We thank the Reviewer for this comment. Τhe current manuscript has been submitted as a ‘’Review’’ and not an ‘’Original Research Paper’’. As disclosed in the introductory part of our manuscript, this review aims to aggregate all available evidence in respect to RV function assessment in severe TR. As clearly explained there is a gap of robust suggestions by the respective guidelines, as to which RV function parameter should be used in clinical practice to define RV dysfunction and inform therapeutic decisions in severe TR. Hence, this review covers the topic by providing a detailed description of advantages, disadvantages and most important studies for each RV function parameter. We do not propose a novel method of RV function assessment as such in this work.  The articles suggested by the Reviewer refer to deep learning methods and neural networks in echocardiography, and do not relate to the topic we discuss. In this manuscript we do not discuss novel methods of echocardiography but rather refer to the assessment of RV function in clinical practice. Such methods have not been used in clinical practice and far outreach the scope of this review.

Point 2: The advantage of the proposed method with respect to other methods in the literature should be clarified.

Response 2: We thank the Reviewer for this comment. As stated above (please see Point 1) in this review we do not propose a novel echocardiographic method, but rather discuss the available evidence of methods in current practice. We conclude that strain imaging might be the best available method to approach RV function in severe TR and provide a detailed discussion of its advantages (see Page 9 of manuscript, example: RV longitudinal strain offers substantial benefits over traditional indices of RV longitudinal function such as TAPSE or S’ as they are less dependent on geometry, cardiac translational motion and loading conditions; RV longitudinal strain provides information on the myocardial contraction beyond a mere description of volumetric RV changes etc.)

Point 3: The paper does not provide significant experimental details needed to correctly assess its contribution: What is the validation procedure used?

Response 3: We thank the Reviewer for this comment. Τhe current manuscript has been submitted as a ‘’Review’’ and not an ‘’Original Research Paper’’. In this manuscript we do not describe new methods or results where validation procedure would be applicable.

Point 4: Quality of figures is so important too. Please provide some high-resolution figures. Some figures have a poor resolution

Response 4: We thank the Reviewer for this comment. We have now uploaded our figures as TIFF 300dpi (high quality).

Point 5: Conclusion should state scope for future work.

Response 5: We thank the Reviewer for this comment. As part of the conclusion we state that ‘’Although evidence is promising, further prospective research is crucial before strain-driven patient management in TR can be advocated.’’ By this sentence we emphasize that future work should focus on the role of RV strain in severe TR, as it appears to be the most promising method of RV function assessment in TR, based on the available evidence.

Point 6: Results need more explanations. Additional analysis is required at each experiment to show the its main purpose.

Response 6: We thank the Reviewer for this comment. This manuscript is a ‘’Review’’ and not an ‘’Original Research Article’’. Hence, no results are provided throughout the manuscript. A detailed description of advantages, disadvantages and most important studies of each RV function parameter in severe TR is provided.

 Point 7: Need detailed explanation of the preprocessing steps.

Response 7: We thank the Reviewer for this comment. This manuscript is a ‘’Review’’ and not an ‘’Original Research Article’’. Therefore, this comment is not applicable to our manuscript.

Reviewer 2 Report

Authors present a review of the imaging methods for the evaluation of tricuspid regurgitation. 

They highlight current advantages and limitation of conventional echo parameters (TAPSE, S’, FAC), and about advanced applications (Strain, 3D). Next to them they place the parameters derived from TC (volumes) and CRM (Strain).

The review is complete, the references updated well analyzed. The concluding remarks about the prospect of integrating all methods are also interesting. Some mistakes to check (as stain/strain)

Author Response

Manuscript ID: diagnostics-2514341

Type of manuscript: Review

Title: Unmet Needs in the Assessment of Right Ventricular Function for
Severe Tricuspid Regurgitation

Response to Reviewer 2 Comments

Point 1: Authors present a review of the imaging methods for the evaluation of tricuspid regurgitation. 

They highlight current advantages and limitation of conventional echo parameters (TAPSE, S’, FAC), and about advanced applications (Strain, 3D). Next to them they place the parameters derived from TC (volumes) and CRM (Strain).

The review is complete, the references updated well analyzed. The concluding remarks about the prospect of integrating all methods are also interesting. Some mistakes to check (as stain/strain).

Response 1: We thank the Reviewer for the positive feedback and for the comprehensive overview of our work. Minor spelling mistakes have been corrected throughout the revised manuscript.

Reviewer 3 Report

Anastasiou et al focused on the multi parametric assessment of RV function in the presence of severe TR. They reviewed the novel imaging modalities other than standard echocardiography, like 3D echocardiography, longitudinal strain evaluation, cardiac computed tomography and cardiac magnetic resonance, in order to optimize therapeutic decision-making in this complex patients, starting from what was most recently published on the subject. I only have a few questions:

- the target of their study never emerges in the title, better highlighted in the text: Assessment of Right Ventricular Function in SEVERE Tricuspid Regurgitation. I would also change the Running title: Right ventricular function to SEVERE tricuspid regurgitation. It should be changed;

- it would have been interesting to introduce a discussion of the functional and biochemical aspects in the assessment of right ventricular function;

- caption of figure number 2, page 9:  it lacks the citation of the article from which the authors took the proposed cut-offs;

- Landzaat et al, in a recent meta-analysis on the determination of standard reference values of right ventricular longitudinal systolic strain, suggested that the lower reference values for RVLS in the current recommendations (with a cut-off value of - 20%) is underestimated: what do the authors think about it?

- Ancona et al, recently published an article focusing on the ratio between right ventricular longitudinal strain and pulmonary arterial systolic pressure in patients with severe tricuspid regurgitation. It should be considered in your article;

- on the quality of the draft, the authors often use sentences that are too long, I would ask for shorter sentences with less use of semicolons;

I think that minor editing of English language in sentence structure are required.

Reviewer 4 Report

This review article reported how to evaluate the right ventricle function and what parameters were helpful in clinical settings.  Especially this article showed well that no specific parameter was obtained from echocardiography regarding determining the optimal timing to intervene in the tricuspid regurgitation.

Recently, RV-PA coupling was reported to be a powerful independent predictor of prognosis in heart failure patients (Eur J Heart Fail 2017; 19: 873–879 ).  Moreover, advanced RV-PA coupling, which was calculated as the ratio of RV-free wall longitudinal strain and right ventricle systolic pressure, was reported to be a predictor in patients with heart failure and secondary mitral regurgitation (JACC Interv, 2021; 14. 2231-2242).  RV-PA coupling was also reported to have an inverse relation with all-cause mortality among TR patients (JACC Interv, 2022; 5. 462-464).  It will be better to add this information for readers of this journal. 
